# Management of Patients with Severe Asthma and Chronic Rhinosinusitis with Nasal Polyps: A Multidisciplinary Shared Approach

**DOI:** 10.3390/jpm12071096

**Published:** 2022-07-01

**Authors:** Veronica Seccia, Maria D’Amato, Giulia Scioscia, Diego Bagnasco, Fabiano Di Marco, Gianluca Fadda, Francesco Menzella, Ernesto Pasquini, Girolamo Pelaia, Eugenio Tremante, Eugenio De Corso, Matteo Bonini

**Affiliations:** 1Otolaryngology, Audiology and Phoniatric Operative Unit, Department of Surgical, Medical, Molecular Pathology and Critical Care Medicine, Pisa University Hospital, 56124 Pisa, Italy; veronicaseccia@gmail.com; 2UOSD Malattie Respiratorie “Federico II”, Ospedale Monaldi, AO Dei Colli, 80131 Napoli, Italy; marielladam@hotmail.it; 3Department of Medical and Surgical Sciences, University of Foggia—University Hospital, Policlinico Riuniti di Foggia, 71122 Foggia, Italy; giuliascioscia84@gmail.com; 4Allergy and Respiratory Diseases, Department of Internal Medicine (DIMI), IRCCS Policlinico San Martino, University of Genoa, 16132 Genoa, Italy; dott.diegobagnasco@gmail.com; 5Department of Health Sciences, University of Milan, Pneumology, ASST Papa Giovanni XXIII, 24127 Bergamo, Italy; fabiano.dimarco@unimi.it; 6Department of Otolaryngology, San Luigi Gonzaga Hospital, University of Turin, 10043 Orbassano, Italy; dott.fadda@gmail.com; 7UOC Pneumologia, Ospedale “S. Valentino”–AULSS 2 Marca Trevigiana, 31044 Montebelluna, Italy; francesco.menzella@aulss2.veneto.it; 8ENT Division “Bellaria Hospital”, AUSL, 40139 Bologna, Italy; pasquiniernesto@gmail.com; 9Dipartimento di Scienze della Salute, Università Magna Graecia, 88100 Catanzaro, Italy; pelaia@unicz.it; 10Otolaryngology Head and Neck Surgery Unit, A.O.R.N. Ospedali dei Colli, 80131 Napoli, Italy; eugeniotremante960@gmail.com; 11Department of Otolaryngology Head and Neck Surgery, Fondazione Policlinico Universitario A. Gemelli IRCSS, Università Cattolica Sacro Cuore, 00168 Rome, Italy; 12Department of Cardiovascular and Thoracic Sciences, Università Cattolica Sacro Cuore, 00168 Rome, Italy; matteo.bonini@unicatt.it; 13Department of Clinical and Surgical Sciences, Fondazione Policlinico Universitario Agostino Gemelli—IRCCS, 00168 Rome, Italy; 14National Heart and Lung Institute (NHLI), Imperial College London, London SW3 6LY, UK

**Keywords:** asthma, chronic rhinosinusitis, nasal polyps, CRSwNP, biologics, patient-reported outcomes, cytology, surgery, type-2 inflammation, multidisciplinary, precision medicine

## Abstract

Chronic rhinosinusitis (CRS) is one of the most frequent comorbidities associated with asthma and it contributes to an amplified global disease burden in asthmatics. CRS prevalence is much higher in asthmatic patients compared to the general population and it is more frequently related to severe asthma, especially in presence of nasal polyps (chronic rhinosinusitis with nasal polyps, CRSwNP). Moreover, asthma exacerbation has a higher occurrence in CRSwNP. From a pathologic point of view, CRS and asthma share similar and connected mechanisms (e.g., type-2 inflammation). A multidisciplinary approach represents a crucial aspect for the optimal management of patients with concomitant asthma and CRSwNP and improvement of patient quality of life. An Italian panel of clinicians with different clinical expertise (pulmonologists, ear, nose and throat specialists, immunologists and allergy physicians) identified three different profiles of patients with coexisting asthma and nasal symptoms and discussed the specific tracks to guide a comprehensive approach to their diagnostic and therapeutic management. Currently available biological agents for the treatment of severe asthma act either on eosinophil-centered signaling network or type-2 inflammation, resulting to be effective also in CRSwNP and representing a valid option for patients with concomitant conditions.

## 1. Introduction

The European Position Paper on Rhinosinusitis and Nasal Polyps (EPOS 2020) [1], the International Consensus Statement on Allergy and Rhinology (ICAR-RS-2021) [2], and the European Forum for Research and Education in Allergy and Airway Diseases (EUFOREA) [3] emphasized the burden on subject suffering from chronic rhinosinusitis (CRS) characterized by symptoms not only linked to sino-nasal mucosa inflammation, but also the extra-sinus problems on sleep, cognitive function, fatigue, depression and general wellbeing. This burden is also reflected in healthcare costs and loss of productivity [1,2,3].

Literature evidence, including epidemiological and clinical studies, strongly indicate that CRS and asthma are strictly connected and could co-occur, especially in presence of nasal polyps. The prevalence of CRS with nasal polyps (CRSwNP) in patients with asthma may vary according to the disease severity. In fact, CRSwNP and severe asthma are more frequently associated (57.1–62%) compared with mild asthma (38–42.9%) [4,5]. Furthermore, the prevalence of CRSwNP in severe asthma patients is variable across country regions [6,7]. 

Literature data confirm that severe asthma in the presence of CRSwNP is more difficult to treat and control, due to increased airway obstruction and higher number of exacerbations [4,5].

### 1.1. CRSwNP and Asthma: Common Pathophysiological Mechanisms

In recent years, it has been highlighted a correlation between type-2 inflammation and different manifestations of CRSwNP with severe asthma [6]. Indeed, in more than half of all severe asthmatics and the majority of CRSwNP cases, type-2 inflammation has been detected [6,7].

In this regard, the existence of a pathophysiological continuum of eosinophilic inflammation between lower and upper airways has been suggested, usually described under the umbrella name of “united airway disease” theory [5,8,9].

In this setting, eosinophil-predominant inflammation is driven by different type-2 cytokines (mainly IL-4, IL-13 and IL-5) and can be activated by multiple inflammatory triggers targeting the epithelium, such as allergens, superantigens and pathogens [8]. In particular, primary CRS is characterized by both type-2 and non-type-2 endotype, while pathological processes (e.g., inflammatory, mechanical or immunological) are at the basis of secondary CRS [10]. Interestingly, a recent study compared modifications of the nasal mucosa in both COVID-19 and CRSwNP patients and showed a different tissue interleukin IL-33 concentration, resulting higher in CRSwNP subjects thus suggesting a different inflammatory pattern [11].

However, immunologic profile is more complex, considering that different endotypes may coexist in the same subject. For instance, CRSwNP is mainly associated with non-eosinophilic inflammation involving Th1/Th17 pathways, while the expression of type-2 cytokine is prevalent in Chronic Rhinosinusitis Without Nasal Polyps (CRSsNP) [12,13].

### 1.2. Multidisciplinary Approach: A Crucial Point in Comorbidities Management

Different specialists should be required for the clinical management of patients with CRSwNP and asthma: allergists, pulmonologists, ear, nose and throat (ENT) specialists, clinical immunologists.

Consequently, their knowledge of the pathogenic mechanisms and the new treatment options is essential to promote a common approach, in particular when considering monoclonal antibodies (mAb) therapy in case of difficult to treat CRSwNP [8,12,13,14].

From diagnosis to treatment strategy definition and patient follow-up, a multidisciplinary approach is essential to obtain the maximum clinical benefit and to improve patient quality of life (QoL).

### 1.3. Indications for Biological Treatment in Severe Asthma and CRSwNP

Recently, a wider knowledge regarding the pathogenic mechanisms involved in the disease led to the development of several biologics (monoclonal antibodies; mAbs) for severe asthma treatment. These biological treatments, acting on the type-2 inflammation and eosinophil-centered signaling network, have been shown effective also in CRSwNP.

Five mAbs are currently approved for severe uncontrolled asthma (Table 1).

For the treatment of CRSwNP patients, the latest EPOS has defined the criteria for the use of biological agents; in particular, subjects with diffuse bilateral nasal polyposis underwent endoscopic sinus surgery (ESS) or patients not fitting for surgery are eligible for biological therapy if at least 3 criteria are satisfied, as reported in Figure 1 below [1]:

Therefore, therapy with biologics for nasal polyposis should be implemented only after respecting a growing gradient of both pharmacological and surgical intervention [8]. The same rationale is also adopted in the GINA guidelines for treatment of severe asthmatic patients with biologics [21].

## 2. Materials and Methods

This paper highlights three different profiles of patients affected by severe asthma and concomitant CRSwNP, and the correspondent flows to guide their specific management. The management tracks have been conceived to ensure that the most effective diagnostic path and therapeutic option are chosen, and the patients can achieve the best outcomes in terms of both asthma and nasal symptoms control.

Four ENT specialists and eight asthma specialists (immunologist, pulmonologist and allergologists) were selected to form a scientific multidisciplinary panel. These experts were identified because of their broad knowledge in treating and managing severe asthma patients and comorbid CRSwNP, in the clinical practice. In addition, authors belong to Asthma Centers which currently implement a structured multidisciplinary approach for the management of the above-mentioned patient profile.

The main objectives to be addressed by the panel were: (1) to describe the current management of severe asthma patients with concomitant CRSwNP, (2) to identify distinct clinical profiles within this specific subset of patients and (3) to discuss their optimal diagnostic and therapeutic management. The patient profiles and the management flows have been developed by the members of the board on the basis of their clinical experience, knowledge of scientific literature and in respect of the international guidelines.

Authors were also involved as advisors in two virtual web meetings covered by a drug company (AstraZeneca): the first meeting was focused on the discussion of the current management of patients with severe asthma and nasal symptoms, identifying possible gaps and solutions. After this first meeting, initial and preliminary three management flows for three different profiles of patient with asthma and nasal symptoms have been designed by the group. During the second virtual meeting, the members of the panel were divided into three focus groups each of them comprising both asthma and ENT specialist who discussed and reviewed the initial flows. The output of each focus group was discussed with the whole specialists during a web meeting, in order to finalize the three patient journeys and determine a multidisciplinary shared approach. The following three patient profiles were defined:Patient with asthma who needs to start a biologic therapy being visited at the allergy/pulmonary unit complaining about nasal symptoms.Patient with severe asthma with an ongoing biologic therapy, being visited at the allergy/pulmonary unit complaining about nasal symptoms.Patient with severe CRSwNP being visited at the ENT unit and complaining about asthma symptoms.

## 3. Results and Discussion of Patients’ Management Flows

The three patients’ management flows are reported in this section together with the related discussion.

### 3.1. Patient with Severe Asthma Who Needs to Start a Biologic Therapy at the Allergy/Pulmonary Unit Complaining about Nasal Symptoms

The first case scenario is related to a patient who needs to optimize his asthma treatment eligible for biologic therapy and complains about nasal symptoms during the control visit with pulmonologist (Figure 2).

Obtaining a good symptoms control, reducing the risk of asthma related mortality, exacerbations, persistent respiratory problems and therapy adverse events, all represent the most relevant clinical targets in the long-term asthma management. Patient’s own goals regarding disease and treatment have to be discussed with the physician as well [21].

Therefore, all patient assessments (first visit and follow-up) should include different endpoints (Figure 2): (1) functional outcomes (e.g., measured with spirometry, reversibility test or responsiveness to inhaled methacholine, the study of the small airways); (2) clinical outcomes (e.g., number of exacerbations in the last year); (3) laboratory evaluations (inflammatory parameters e.g., blood eosinophil count, fractional exhaled nitric oxide); (4) individual outcomes, to be evaluated through the collection of patient-reported outcomes (PROs) in order to better characterize the patient and his perception of the disease [21].

These parameters could also contribute to assess the disease evolution over time. Ongoing therapies, such as ICS/LABA (Inhaler Corticosteroids/Long-Acting Beta-Agonists), SCS (Systemic Corticosteroids), as needed SABA (Short-Acting Beta-Agonists) or ICS/LABA use, related adherence and the inhalation technique has to be also investigated. Response to treatment should be reviewed after 3–6 months as per GINA recommendations [21].

Moreover, for difficult to treat asthma and for non-severe asthma, collecting information regarding the presence of comorbidities such as rhinosinusitis, obesity, gastro-esophageal reflux disease is of key importance, even for optimizing both patient and treatment management. Indeed, comorbidities may contribute to respiratory symptoms, impaired QoL and, in some cases, asthma control [21,22]. Therefore, the examination of the upper airways should be arranged for patients with asthma though a referral to ENT specialist in order to detect possible CRS (or CRSwNP), in particular in patients complaining nasal symptoms. In addition, even when a CRS has been previously diagnosed, a multidisciplinary approach (pulmonologist and ENT specialist) is required to assess the asthma concomitant condition and to define the most adequate treatment strategy.

Focusing on Figure 2, point b, it is interesting to highlight that the diagnosis of asthma currently includes for several diseases with distinct endotypes and phenotypes. Regarding asthma endotypes, it is possible to distinguish two main subtypes: type-2 high (high eosinophilic inflammation); type-2 low (with both neutrophilic and paucigranulocytic inflammation, together with steroids resistance) [23].

The identification of the specific endotypes could support the most adequate asthma management considering the related implications in treatment approach and prognosis [24].

In this regard, identifying the presence of type-2 high endotype could guide the evaluation of treatment choice with biologics targeting type-2 pathway cytokines [25]. In addition, specialists suggest that response to steroid therapy may be considered as a marker to guide the choice towards a biological agent [26].

Dominant type-2 response (type-2 cytokines) and eosinophilia (circulating/local IgE) are present in around 80% of CRS patients. As per EPOS guidelines, the presence of type-2 inflammation is established with a tissue eosinophils (EOS) count ≥ 10/high power field (HPF) or blood EOS count ≥ 250/mcL or total IgE ≥ 100 kU/L. Moreover, local eosinophilic infiltration quantity and the intensity of the inflammatory response are strictly linked to disease prognosis and severity [13,27]. Sino-nasal mucosa specimen collection, its storage and processing should be performed in collaboration with histopathologists [13].

Concerning the multidisciplinary approach, pulmonologists could collect some initial information on patient medical history regarding nasal symptoms before the referral to ENT specialist (as suggested in Figure 2, point c) in order to facilitate the diagnostic process or subsequent patient re-evaluations. It is necessary to underline that, as the diagnosis of CRS (and therefore CRSwNP) is mainly performed through symptoms and clinical signs observation, then supplemented with nasal endoscopy (rhinofibroscopy in primis) and computed tomography (CT), these assessments should be performed by ENT specialists [28].

For CRSwNP diagnosis, endoscopy represents a backbone allowing to perform an adequate phenotyping, disease staging and differential diagnosis [29].

To confirm the CRSwNP diagnosis, CT scan is not sufficient if endoscopy is not assessed; anyway, information from CT scan could both support pre-surgery planning and help in preventing complications during surgery by assessing anatomical variations [30]. Furthermore, Hong H et al. suggested that the use of sinus CT scan in CRSwNP patients could potentially provide indication on glucocorticoid-sensitivity [28].

In the evaluation of patients after endoscopic sinus surgery, radiological imaging could be also considered [31]. In this context, Lund-Mackay score is broadly used for radiologic staging of CRS pre- and post-treatment; this method is also associated with disease severity markers, the surgery type and outcome [32]. Moreover, CT scan is able to highlight recurrent/residual nasosinusal disease and possible bony defects caused during a previous surgery [33].

In addition, specialists discussed the possibility to have a facial CT (or CT of paranasal sinuses) performed by pulmonologists supporting the subsequent evaluation by the ENT specialist, who is mainly responsible for the assessment. The panel of specialists agreed that a risk-benefit estimation in performing facial CT has to be always accomplished. Moreover, CT is not a first level evaluation in patient with nasal symptoms, as it allows a radiological diagnosis and not a clinical one. As CT is related to excessive radiation exposure risk, it may not be strictly necessary in the patient discussed here. For these reasons, the specialists recommend avoiding CT as a screening tool for evaluation of concomitant nasal pathologies in patients with asthma.

Regarding medical history, information on previous surgery for nasal polyps (e.g., number and type of interventions) are essential to better define patient clinical picture. Although endoscopic sinus surgery (ESS) may benefit CRSwNP patients, more than 50% of cases have polyp recurrence and around 30% of patients require a revision surgery at some point [34]. The time relapsed since the last surgery is also an important factor in identifying recalcitrant forms, also considering that the time between additional revision surgeries subsequently reduces [35].

Functional endoscopic sinus surgery (FESS) in patients with coexisting CRSwNP and asthma provide a beneficial effect on both diseases improving objective and subjective measurements [36].

Interestingly, a study evaluating the impact of sinus surgery for CRS on asthma control showed that patients with worse preoperative asthma control level could have an additional positive effect from nasal surgery [37]. Moreover, recent evidence suggest how ESS seems to ameliorate asthma control in patients with CRSwNP and concomitant asthma by suppressing type-2 inflammation [38]. Besides, asthma severity is related with the risk of exacerbations in CRSwNP subjects and in patients with asthma functional parameters improve after nasal polyp surgery.

Regarding post FESS recovery, Jeican et al. underlined the therapeutic benefits of mineral and thermal waters in terms of nasal flow improvement, nasal resistance decrease, mucociliary transport time and pathological microbial flora reduction. Patients should be adequately educated by ENT specialist and family doctors in order to understand the efficacy of this type of therapy in the postoperative recovery [39]. In addition, among alternative therapies for CRSwNP, different possible interventions can be considered, including thermal waters, acupuncture, aspirin desensitization.

As already mentioned, patient-reported outcomes (PROs) have to be collected both at baseline and at following visits to rank the severity of nasal symptoms, for example using visual analogue scale (VAS), or to evaluate the impact of symptoms on patients’ QoL. The 22-item Sino-Nasal Outcome Test (SNOT-22) is the currently used method (validated in the literature) to determine disease control level. The maximum score is 110 (greatest disease impact) while 8.9 points refer to the lowest clinically significant difference. A score > 50 usually indicates uncontrolled disease or patient’s quality of life severely impaired by NP. This questionnaire should be self-completed by the patient [14]. Several studies reported a significant QoL outcome improvement after ESS by using SNOT-22. Anyway, the grade of amelioration is variable and depends on many aspects, such as SNOT-22 score at baseline, asthma prevalence and follow-up duration [40].

Besides, specialists agreed that patient should be supported when compiling SNOT-22 as it includes terminologies difficult to understand by patients, although the presence of a doctor during questionnaire compilation may represent a bias. Authors suggested that SNOT-22 could be performed autonomously by patients and each result can be subsequently discussed with physician. Moreover, as per expert opinion, differences in results may be observed when the questionnaire is accomplished during the visit with pulmonologist or ENT specialist, possibly because of changes in patient’s perception.

UPSIT/Sniffing test can also be used to assess the sense of smell by the recognition or not of standard aromas; the peak nasal inspiratory flow (PNIF) is another method to evaluate can be used together with the assessment of polyposis severity and its perceived impact [14].

In conclusion, the multidisciplinary approach and the referral to ENT specialist is crucial for the best management of patients with asthma with nasal symptoms. For patients with severe asthma, the evaluation by ENT specialist should be always required and performed. Several parameters and clinical characteristics have to be examined in order to optimize treatment strategy (summary of the first flow in Figure 3).

### 3.2. Patient with Severe Asthma with Ongoing Biologic Therapy at the Allergy/Pulmonary Unit Complaining about Nasal Symptoms

Considering this specific clinical case, the first step is to verify whether or not asthma is controlled by evaluating several aspects (Figure 4, point a).

In selecting the best asthma therapy, the level of asthma control represents a crucial factor to be considered [40]. First of all, adherence and inhaler technique for the ongoing asthma treatment have to be examined. A possible therapy with systemic corticosteroids should be verified as well, focalizing on the reason of use (e.g., prescribed by ENT specialist for a previous nasal comorbidity).

Patient-reported outcomes (asthma control test, ACT; asthma control questionnaire) could also provide relevant information on asthma control level and QoL impact. The ACT application is suggested in a clinical routine as a useful tool not only to assess disease control and patient outcomes (e.g., symptomology and future risk), but also for a proper management of asthmatics and for identifying alternative treatment approach [41].

Comorbidities, such as eosinophilic granulomatosis with polyangiitis (EGPA), should be checked. Patients with EGPA mainly have severe asthma which often do not respond to immunosuppressive treatments or to corticosteroids and systemic vasculitis could early occurs [42]. Moreover, in patients with EGPA and long-term severe/uncontrolled asthma, pulmonary and upper airways manifestations are frequent even at the baseline [43]. Immunosuppressive drugs could be prescribed for both maintenance therapy and EGPA exacerbation. In addition, active EGPA shows high IL-5 levels suggesting that its inhibition represents a relevant therapeutic target in SEA. Furthermore, monoclonal antibody anti IL-5 and anti IL-5 receptor efficacy has been reported in patients with EGPA and asthma (even in presence of CRSwNP) [42].

Nasal comorbidity has to be always assessed as it could be associated with different features of asthma [44] and the presence of CRSwNP is related with increased asthma exacerbation frequency, representing a predictor of future exacerbations [7]. CRSwNP and comorbid asthma are also associated with poorly controlled asthma [6] Therefore, a patient evaluation by ENT specialist is crucial in order to assess nasal symptoms and to verify related treatment adherence (e.g., corticosteroids). As already discussed in the previous patient profile, SNOT-22 score and visual analogic scale (VAS) questionnaire may be useful in order to evaluate patient QoL together with nasal symptoms impact [14]. Basal scores are essential for future comparison with follow up assessments.

ENT specialist should be also responsible to investigate whether previous surgery has been performed in patients with asthma and nasal symptoms (Figure 4, point d). Independently from the technique of surgery, a recurrent CRSwNP disease could occur at some point, ranging from 4 to 60% (median of 20% in maximum 2 years). In case of nasal polyp recurrence, a revision surgery could be necessary in 4–27% of patients with a follow-up period ranging from 12 and 60 months. Moreover, several evidence from literature reported that CRSwNP patients with asthma or Aspirin-exacerbated respiratory disease (AERD) show higher recurrence rates [45].

In this context, it is clear that patient with severe asthma with ongoing biologic therapy complaining about nasal symptoms requires a continuous multidisciplinary approach, which becomes essential when evaluating a possible biologic treatment switching (Figure 4, point e). As per GINA guidelines, an add-on type-2 targeted biologic has to be considered in case of exacerbations or poor symptom control (although ICS-LABA high dose); this approach is suggested also for patients with allergic or eosinophilic biomarkers or when oral corticosteroids maintenance treatment is required. In case of no response, a switching to a trial of a different type-2 targeted therapy should be considered [22].

In patients with difficult to control asthma, a switch between biologics could be necessary not only for a lack of treatment response but also in case of safety issues, need of a different dosing schedule, patient’s preferences, comorbidities (e.g., CRSwNP), infections or other conditions such as pregnancy [46].

Panel of specialists agreed that, after a multidisciplinary evaluation of a patient with controlled asthma and CRSwNP, switching to a different biologic may be considered also to optimize the use of corticosteroids and avoid possible surgery. The contribution of the ENT specialist is important to investigate the reason of uncontrolled inflammatory state at the nasal level. A change in biologic agent should be carefully evaluated for the potential risk of asthma control loss.

For instance, a switch from an anti-IL-5 (or anti-IL-5Rα) to anti-IL-4-R-alfa biologic might be evaluated for each patient with a personalized approach not only to reach the control of severe asthma and oral corticosteroid tapering, but also to limit possible eosinophilic complications [47].

To conclude, as several aspects must be investigated in patients with asthma under biologic treatment and with nasal symptoms, a close collaboration between pulmonologist, ENT specialist, immunologist is required not only to define the best treatment approach (including biologic switching) in order to obtain both CRSwNP and asthma control, but also to set a multiparametric management strategy with a perspective vision (summary of the second flow in Figure 5).

### 3.3. Patient with Severe CRSwNP at the ENT Unit Complaining about Asthma Symptoms

In patients presenting with CRSwNP, a confirmation of this diagnosis has to be firstly assessed by ENT specialist considering several parameters, as indicated in Figure 6, point a.

As suggested by recent position papers, the evidence of type-2 inflammation should be high likely in these patients and for this reason, searching for local or systemic eosinophils and IgE count is advised [11,47,48] Considering that other biomarkers currently used to define type-2 disease are blood eosinophilia, IgE levels, EPOS group suggested specific cut offs for the following biomarkers: >250/µL for blood eosinophilia and >100 kU/L for total IgE [1,12].

The amount of local eosinophilic infiltration and the overall intensity of the inflammatory response are closely related to the prognosis and severity of nasal disease [26]. The most common used techniques to define local inflammation include nasal biopsy, nasal brushing or scraping (nasal cytology), nasal lavage fluid and nasal suctioning of secretions. The diagnosis of eosinophilic CRS requires quantification of the numbers of eosinophils, through the analysis of at least three of the densest collections of eosinophils (very rich fields) in the samples counted at hpf (~400×). The EPOS steering group specified that the minimal cut-off to achieve evidence of type-2 inflammation on tissue samples was eosinophils > 10/hpf.

Other aspects to be underlined are the wide range of CRSwNP manifestations, the individual variability of its severity and the related disease impact on QoL. Therefore, the possibility to measure and better define CRSwNP is crucial. The EUFOREA group defined Severe CRSwNP as “bilateral CRSwNP with a nasal polyp score (NPS) of at least 4 of 8 points and persistent symptoms, including loss of smell and/or taste, nasal obstruction, secretion and/or postnasal drip, and facial pain or pressure, with the need for add-on treatment to supplement intranasal corticosteroids” [3,49].

Besides, validated QoL markers such as VAS and SNOT-22 have been currently adopted to define severe CRSwNP [8]. In addition to these two PROs, UPSIT/Sniffing test is useful to assess the effect of nasal polyposis symptoms on sense of smell and patients’ QoL (Figure 6a) [14].

Corticosteroid treatment is another aspect to be evaluated (Figure 6a,c). For instance, for uncontrolled severe CRSwNP receiving inhaled corticosteroids (long term) together with 7–10-day bursts of systemic corticosteroids and never underwent to surgery, ESS could be considered as the first line treatment (if feasible as per ENT specialist opinion).

Moreover, panel of specialists highlight that endocrinologist should be included in the multidisciplinary team to assess possible symptoms or damage associated with corticosteroid treatment (e.g., cortisol presence in serum or urine).

In case of comorbidities, ENT specialist should involve a pulmonologist in patients’ assessment in order to confirm asthma diagnosis and related phenotype [15]. Panel of specialists agreed that ENT specialist could perform an initial “pre-screening” of patient by collecting information regarding treatment for respiratory disease (e.g., SABA, INCS/LABA use) and PROs (such as ACT and ACQ-6) (Figure 6a), which may be useful to subsequent patient evaluation by pulmonologist.

Through a multidisciplinary approach the most adequate treatment (including biologic options) can be then defined. If the patient is diagnosed with severe asthma and biologic is considered the most adequate option for severe asthma and concomitant CRSwNP, asthma specialists should be principally responsible for the management of this treatment approach, while the role of the ENT specialist is essential for patients with severe uncontrolled CRSwNP without asthma or with mild-to-moderate controlled asthma [12].

In managing patients’ comorbidities, surgery may provide a relief of sino-nasal symptoms and may ameliorate asthma control. However, surgery should be postponed while evaluating the biological therapy efficacy on sino-nasal symptoms and in reducing nasal polyp score. For patients with this complex clinical profile, the cooperation between different specialists is recommended, in particular during treatment to verify its efficacy on both asthma and CRSwNP.

In addition, for patient with CRSwNP and concomitant conditions potentially related to asthma (e.g., obesity, gastro-esophageal reflux disease—GERD; obstructive sleep apnea syndrome—OSAS), specialists strongly suggest a patient referral to pulmonologist. Other comorbidities, such as NSAID-exacerbated respiratory disease (NERD), should be investigated as may affect both asthma and nasal polyposis.

Similarly to the previous described patient profiles, also for a patient with CRSwNP and concomitant asthma, the history of previous surgery was identified as a crucial element in terms of timing of recurrence and symptoms control duration after surgery (Figure 6a,c). Recent studies reported a high risk of treatment failure (need for further surgery) in patients presenting symptomatic CRSwNP recurrence within 3 years from surgery [32].

A careful evaluation should be performed by surgeons on the need of a revision surgery making a distinction based on the time factor: new procedure required after a short period from the first surgery or revision needed after several years with a well-controlled disease. Moreover, it is essential to involve patient into the decision process (surgery vs. switch to biologic) [12].

In conclusion, the complexity of patient with CRSwNP and comorbid asthma has to be adequately managed through specialist close collaboration. Therefore, symptoms suggesting a possible presence of asthma should be deeply investigated by pulmonologist, while the global management strategy has to be established with a multidisciplinary approach over time (summary of the third flow in Figure 7).

## 4. Final Considerations

In patients with asthma, multiple comorbidities might overlap with asthma symptoms, lower adherence to treatment, interfere with the response to asthma medications, and affect the quality of life (QoL). CRSwNP commonly coexists with type-2 high asthma, contributing to the overall disease burden and the loss of asthma control.

The identification of inflammatory phenotypes of the patient, and therefore the differentiation between type-2 versus non type-2 inflammation, is supportive in determining eligibility for targeted biologic therapies in asthma and in comorbid CRS.

The management of patients where asthma and CRSwNP coexist requires a multidisciplinary approach by at least ENT specialist, immunologist and pulmonologist in order to evaluate symptoms and clinical history, confirm diagnoses and to identify the best treatment strategy aimed at controlling both diseases and preventing clinical exacerbations.

This approach can possibly detect patients who may require treatment with biologics even at an earlier stage in the disease process.

In this document, three patient profiles (presenting asthma and nasal symptoms) have been above discussed by the multidisciplinary panel of Italian specialists and the most relevant observations can be highlighted as follows:A close collaboration between pulmonologist, ENT specialist and allergist/immunologist is required for patients with asthma complaining nasal symptoms and vice versa, since the first patient take-charge when asthma or CRS diagnosis has to be done. These professionals should be working within the same center or in close collaboration with a multidimensional network.Patient reported outcomes are useful tools for patient QoL assessment and can be recorded by different specialists to allow collection and tracking changes in clinical symptoms over time and allowing better disease control among patients and specialists.Comorbidities should be always investigated as they affect both asthma and CRSwNP control and play a relevant role in patients’ characterization and in treatment selection.Information on previous and ongoing treatments (drug for asthma and/or corticosteroids also for nasal symptoms) have to be collected in terms of reason of use, frequency, adherence, inhalation technique (depending on type of therapy) as medical history can often direct care.Medical history should always include surgery (number and type of interventions; time from last surgery and time to recurrence).Nasal inflammation has to be carefully examined. Cytology and tissue eosinophilia can provide relevant and accurate information on patient condition; the detection of nasal eosinophilic inflammation represents an early marker for identification of a more aggressive inflammatory phenotype in patients with CRSwNP.Monoclonal antibodies have been demonstrated to be very useful in the management of chronic eosinophilic diseases such as asthma and are demonstrating effective results also in type-2 inflammatory CRSwNP. The benefit of biological therapies (with the relative clinical improvements) should be evaluated by a careful measurement of the patient’s multidimensionality, therefore not only considering pulmonary or nasal conditions but applying an integrated approach. Likewise, the limits of efficacy of biological therapy might be verified when patients are evaluated with a multidisciplinary collaboration.

As final key message, considering the complexity of clinical manifestations and pathophysiological mechanisms underlying concomitant asthma and CRSwNP, multidisciplinary collaboration is essential to achieve an overall positive effect on clinical course of both diseases, related psychological impact on patients and on burden healthcare service.

## Figures and Tables

**Figure 1 jpm-12-01096-f001:**
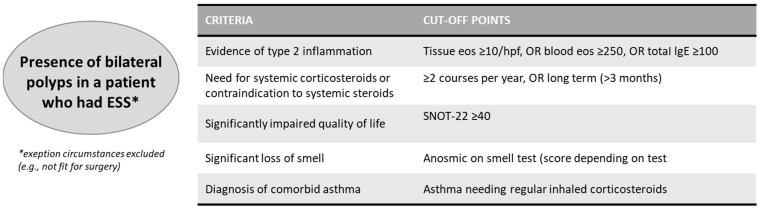
Criteria for biological treatment in CRSwNP. Eos: eosinophils; Hpf: high power field; SNOT-22: 22-item Sino-Nasal Outcome. Adapted from EPOS 2020.

**Figure 2 jpm-12-01096-f002:**
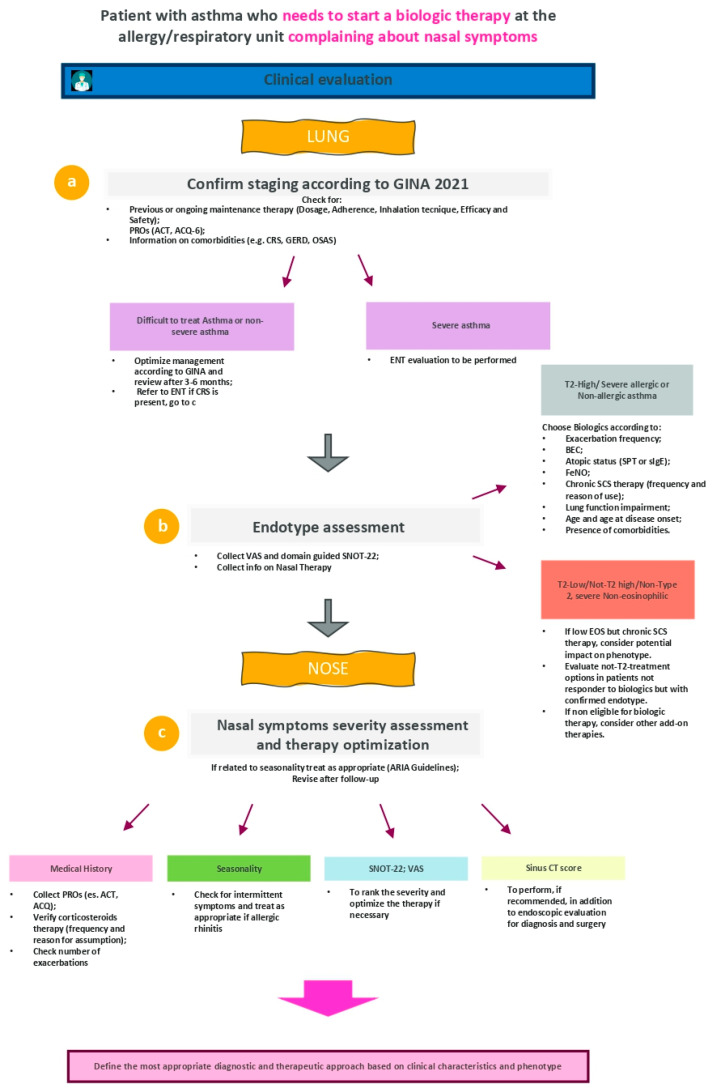
Detailed flow for the management of a patient with asthma who needs to start a biologic therapy at the allergy/pulmonary unit complaining about nasal symptoms. GINA = Global Initiative for Asthma; ENT = ear, nose and throat specialist; EOS = eosinophil count; BEC = blood eosinophil count; SPT = Skin prick test; sIgE = serum immunoglobulin E; FeNO = fractional exhaled nitric oxide; SCS = systemic corticosteroids; T2 = type-2; CRS = chronic rhinosinusitis; INCS = intra nasal corticosteroids; VAS = visual analogic scale (for nasal symptoms); SNOT-22 = sino-nasal outcome test on 22 items; PROs = patient-reported outcomes.

**Figure 3 jpm-12-01096-f003:**
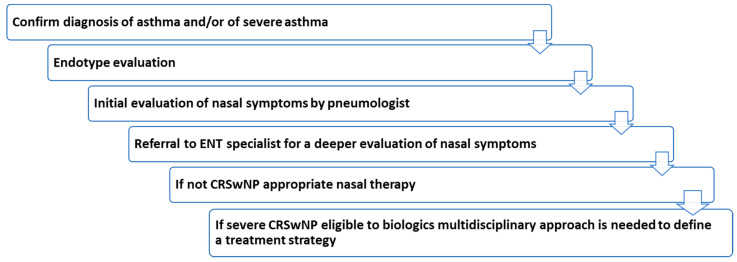
Summary of the first flow. ENT = otolaryngology; CRSwNP = chronic rhinosinusitis with nasal polyps.

**Figure 4 jpm-12-01096-f004:**
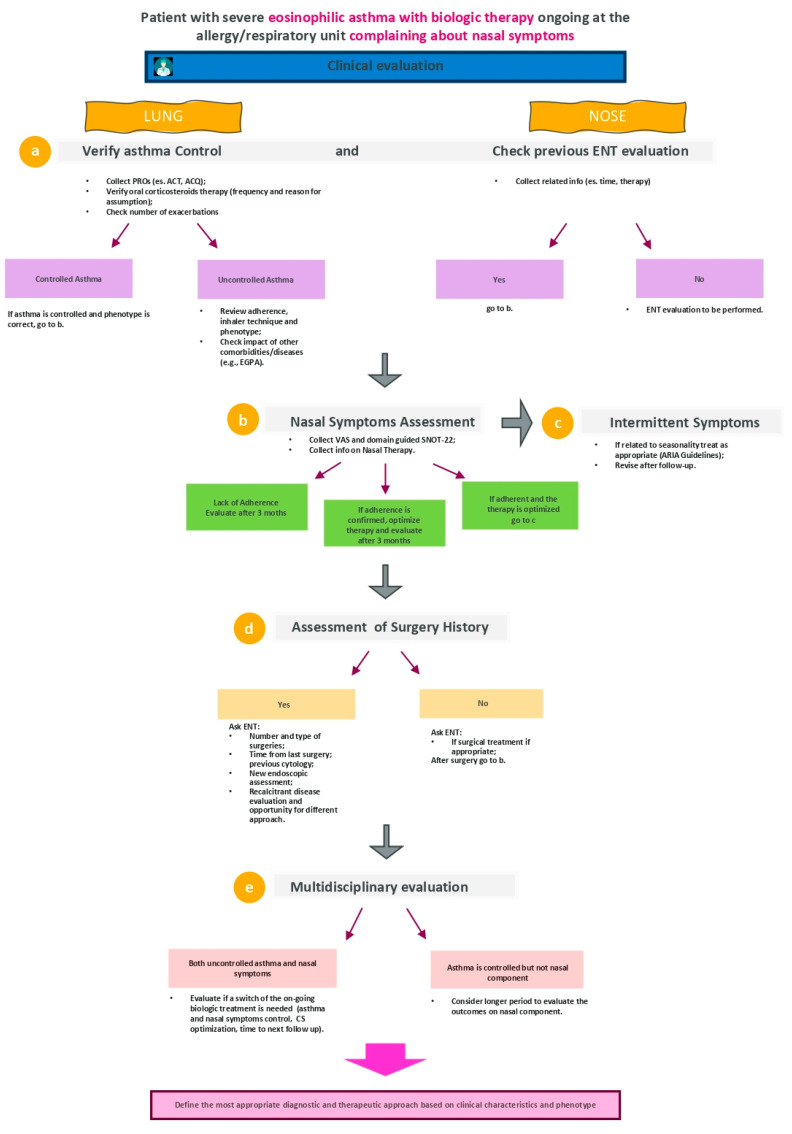
Flow for the management of a patient with severe eosinophilic asthma with ongoing biologic therapy at the allergy/pulmonary unit complaining about nasal symptoms. EGPA = eosinophilic granulomatosis with polyangiitis; VAS = visual analogic scale; SNOT-22 = sino-nasal outcome test on 22 items; ENT = ear, nose and throat specialist; ARIA = Allergic Rhinitis and its Impact on Asthma; PROs = patient-reported outcomes; ACT = asthma control test; ACQ = asthma control questionnaire; CS = corticosteroids.

**Figure 5 jpm-12-01096-f005:**
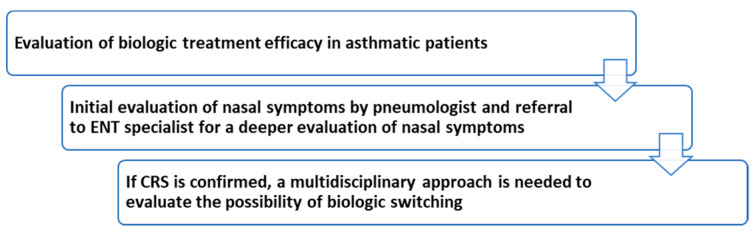
Summary of the second flow. ENT = otolaryngology; CRS = chronic rhinosinusitis.

**Figure 6 jpm-12-01096-f006:**
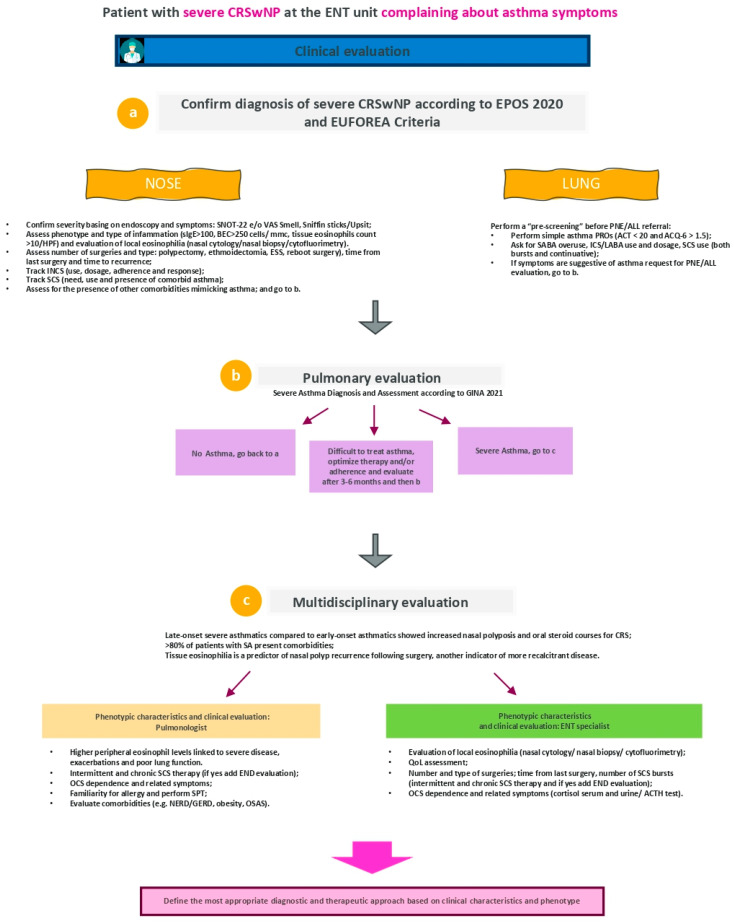
CRSwNP = chronic rhinosinusitis with nasal polyps; EPOS = European Position Paper on Rhinosinusitis and Nasal Polyps; SNOT-22 = sino-nasal outcome test on 22 items; NPS = nasal polyps score; sIgE = serum immunoglobulin E; BEC = blood eosinophil count; HPF = high power field; SCS = systemic corticosteroids; GERD = gastro-esophageal reflux disease; OSAS = obstructive sleep apnea syndrome; SABA = short-acting beta-2 agonists; PROs = patient-reported outcomes; ACT = asthma control test; ACQ-6 = asthma control questionnaire on 6 items; GINA = Global Initiative for Asthma; SPT = skin prick test; CRS = chronic rhinosinusitis; PNE = pulmonologist; INCS = inhaled corticosteroids; NERD = Non-erosive reflux disease; GERD = gastroesophageal reflux disease; OSAS = obstructive sleep apnea syndrome; END = endocrinologist.

**Figure 7 jpm-12-01096-f007:**
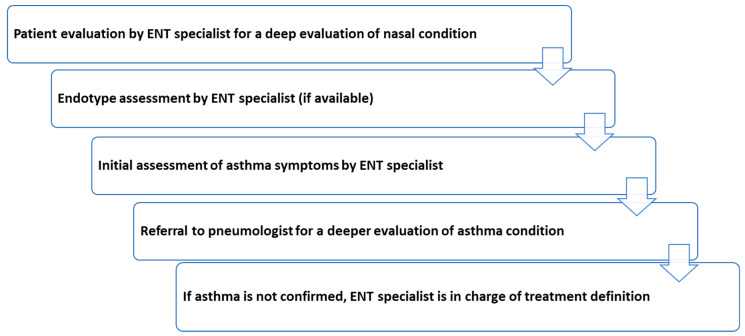
Summary of the third flow. ENT = otolaryngology.

**Table 1 jpm-12-01096-t001:** Biologics currently approved for the treatment of severe uncontrolled asthma. (TSLP = thymic stromal lymphopoietin, EoE Eosinophilic Esophagitis; EG/EGE: Eosinophilic Gastritis/Eosinophilic GastroEnteritis; COPD: Chronic Obstructive Pulmonary Disease; BP: Bullous pemphigoid; HES: Hypereosinophilic syndrome; EGPA: Eosinophilic Granulomatosis with Polyangiitis; NCFB: Non-Cystic Fibrosis Bronchiectasis; CSU: Chronic Spontaneous Urticaria). ^§^ www.clinicaltrials.gov (accessed on 27 June 2022).

Monoclonal Antibody	Omalizumab [15]	Mepolizumab [16]	Reslizumab [17]	Benralizumab [18]	Dupilumab [19]	Tezepelumab [20]
Target	IgE	IL-5	IL-5	IL-5Rα	IL-4Rα, IL-13Rα	TSLP
Route of Administration and dosage related to approved indications	Subcutaneous injection every 2–4 weeks dosing and frequency level determined by serum total IgE and body weight.	Subcutaneous injections 100 mg monthly (SA, CRSwNP) 300 mg monthly HES, EGPA)	Intravenous injection 3 mg/kg every 4 weeks	Subcutaneous injection 30 mg once every 4 weeks for the first 3 doses, then subsequently once every 8 weeks	Subcutaneous injection, 400 mg then 200 mg every 2 weeks (AS) 600 mg then 300 mg every 2 weeks AS and OCS or AS and comorbidity (CRSwNP, AD) 600 mg then 300 mg every 2 weeks CRSwNP, AD	Subcutaneous injection 210 mg monthly
Currently approved indications	Severe allergic asthma CRSwNP Chronic Idiopathic Urticaria	Severe eosinophilic asthma CRSwNP HES EGPA	Severe eosinophilic asthma	Severe eosinophilic asthma	Severe allergic and eosinophilic asthma CRSwNP Atopic dermatitis EoE (FDA)	Severe asthma (FDA)
Other indications under evaluation ^§^	Food allergy	COPD	N/A	CRSwNP EoE HES EGPA COPD EG/EGE NCFB CSU BP Atopic dermatitis	CRSsNP COPD Chronic pruritis Prurigo nodularis BP CSU Chronic inducible cold urticaria Allergic fungal rhinosinusitis Peanut allergy	CRSwNP CSU COPD EoE

## Data Availability

Not applicable.

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
