# Peer review of "Management of Patients with Severe Asthma and Chronic Rhinosinusitis with Nasal Polyps: A Multidisciplinary Shared Approach"

_jpm, 2022, doi:10.3390/jpm12071096_

Round 1

Reviewer 1 Report

Seccia et al propose an interesting study about management of patients with severe 1 asthma and chronic rhinosinusitis with nasal polyps, with a special and original design. I appreciate the fact that the article is complex, but well structured.

Line 87, I recommend the recent study and please cite https://www.ncbi.nlm.nih.gov/pmc/articles/PMC8468618/

Line 185: An Italian panel of experts... Please mention the total number of experts, and number/specialty. Also, please note inclusion criteria or as you defined the expert.

Also, please insert a short discussion about mineral and thermal waters in the management of patients with asthma and CRSwNP (please study and cite http://bioclima.ro/Balneo421.pdf).

Final considerations can be better synthesized.

Author Response

Dear Editor,

Dear Reviewers,

Thank you for your comments and suggestions. All of them guided us to ameliorate the article overall, providing additional details and better balancing contents, with particular reference to biological treatments.

We are also glad that the main objective of our work has been appreciated and that the proposed workflows have been considered helpful manual for physicians to assess patients with asthma and chronic rhinosinusitis with nasal polyps, define a personalized treatment, and select candidates for biological therapy as the multidisciplinary approach is essential for the appropriate management of the three different patient profiles that we have considered.

All authors contributed to the revision and to generate replays to reviewers’ observations.

REVIEWER 1

  1. Line 87, I recommend the recent study and please cite https://www.ncbi.nlm.nih.gov/pmc/articles/PMC8468618/

Replay: Thank you for this interesting reference which was included in the paper (Line 93-96).

  1. Line 185: An Italian panel of experts... Please mention the total number of experts, and number/specialty. Also, please note inclusion criteria or as you defined the expert.

Replay: More details regarding specialists’ selection have been reported in paragraph 2 “Materials and Methods”. Specifically, experts were identified because of their broad knowledge in treating and managing severe asthma patients and comorbid CRSwNP, in the clinical practice. They also belong to Asthma Center where the multidisciplinary approach is already applied.

  1. Also, please insert a short discussion about mineral and thermal waters in the management of patients with asthma and CRSwNP (please study and cite http://bioclima.ro/Balneo421.pdf). 

Replay: Thank you for this interesting reference which was cited in the paper (Line 288-294). Specifically, it was mentioned that mineral and thermal waters can be considered between other possible alternative treatments for patients with asthma and CRSwNP.

  1. Final considerations can be better synthesized.

Replay: final considerations are aimed at summarizing the most relevant observations from the discussions of the three patient profiles. Authors intended to report those clinical aspects and approaches common to all flows and therefore representing key management indications for patients with CRSwNP and asthma.

Reviewer 2 Report

Dear Editor,

Dear Veronica Seccia and colleagues,

The current manuscript aims to provide a helpful manual for physicians to assess patients with asthma and chronic rhinosinusitis with nasal polyps, define a personalized treatment, and select candidates for biological therapy. As a physician, being responsible for such a patient population, I utterly understand the importance of a multidisciplinary approach with a consensus-based workflow.

General comments:

This expert opinion is marked as a review article, however international guidelines reshaped to the local needs and opportunities should include personal observations to reinforce the consensus made by the contributors. Some of the self-cited articles demonstrate the authors’ experience, however, the materials and methods section does not highlight the research process. Whether it is a review (the process of article / guideline / literature screening e.g. PRISMA guidelines for systematic reviews and meta-analyses) or / and reasearch-based (local studies, observations, trials) expert opinion, a more detailed workflow of negotiation is welcome. The „two meetings” is a sort of vague and superficial description of the decision-making. On the other hand, the number of specialists in their field should be indicated.

I ask the authors to provide a table containing all antibody-based medications available for asthma and CRSwNP, highlighting if a given drug is authorized for any or both conditions in the focus of this paper.

Nota Bene, that not all of the mentioned medications have a primary indication when CRSwNP is present. This is important when a patient’s assesment starts in the ENT office. It is confusing that, in the „Indications for biological treatment in CRSwNP” the medication list refers to those are available for asthma, however dupilumab is a medication that can be prescribed for both conditions as a primary choice of biological treatments. Dupilumab is not even mentioned the „Indications for biologics for patients with CRSwNP and asthma” section, where benrolizumab is overrepresented (ANDHI and OSTRO trials are also sponsored by AstrZeneca). This phenomenon raises concern when benrolizumab (Fasenra®), is a product of AstraZenca, which plays a funding role in this publication - as it is declared, AstraZeneca covered the copyediting expenses and is willing to support the publication expenses. This is also suspicious when the authors have nice real-life experiences with the competing drug (https://doi.org/10.3390/jcm11102684), but not cited in the current article. These issues must be solved by more transparent declarations, avoiding presenting results with a single drug. An expert opinion cannot be accepted when a medication is overrepresented (especially when it is not authorized as a biological agent for CRSwNP, but for asthma with an add-on effect – according to the recent negotiation between the FDA and AstraZeneca).

Copyediting must be repeated – grammatical errors, incongruent abbreviations and expressions scattered in the manuscript must be corrected.

The many differences between abbreviation use and the style of sections raise the possibility, that each chapter is written by a different contributor but the senior author didn’t format them to a similar style.

The foremost strength of the article is the spectacular flowcharts – diagrams, however, in the .pdf available for review, they are difficult to read sometimes (e.g. Figure 4 – red text-boxes; vector-based images are welcome for the article to be embedded into the .pdf). I also miss, what is partially included in the text, that at an endpoint of a flowchart, what factors should be evaluated for the most personalized treatment when choosing the monoclonal antibody.

Specific / detailed comments (not all typos or copyediting issues are highlighted):

Line 27 -31: This paragraph must be re-edited. Word repetition, similar style sentences

Line 30: CRSwNP stands for chronic rhinosinusitis with nasal polyps. Introduce the abbreviation when this expression occurs for the first time.

Line 32: „Form” correct to „From”

Line 33: „type-2 inflammation” should be used in this form elsewhere if introduced in this structure.  Insert dash where it is missing

Line 51: update citation: https://doi.org/10.1002/alr.22741

Line 46-51: I also recommend including the EUOPHERA statement here https://doi.org/10.1111/all.1387

Line 85: „T2-inflammation” new term

Line 94: „type 2 or non-type 2” use dash uniformly throughout the article

Line 100: „CRSsNP” has not been introduced

Line 100: „Type 2 cytokine” – use dash and upper-/lower-case letters uniformly throughout the article

Line 103: „Due to the overlapping clinical nature of CRSwNP, conditions [5].” this sentence has no meaning without subject, predicate, object

Line 109: „mAb” has not been introduced before

Figure 1 and Line 130: units missing from the table. SNOT-22 abbreviation must be explained

Line 140: „type 2-targeted” with or without dash?

Line 165: dupilumab can also be prescribed for severe asthma. Why do the authors leave it out from this chapter?

Line 185-188: number of experts and their speciality involved in the study. Selection criteria? A more detailed method description should be included as it is written above

Line 195-200: figures and tables must be placed where they are cited and numbered in the order of their citations.

Line 223: „ICS/LABA, SCS, as needed SABA or ICS/LABA” abbreviations must be explained the first time.

Line 233: „ear, ENT” delete ear

Line 235: „pulmonologist, and ENT specialist” no need for a comma

Line 242-243: „T2-high … a T2-low” again, another form of the abbreviation

Line 248: „FeNO” name it as fractional exhaled nitric oxide and introduce the abbreviation

Line 250: T2

Line 251: T2

Line 256: „type 2” with or without dash ?

Line 261: T2

Line 352: include all abbreviations of the figure e.g. GERD, OSAS

Line 368: the figure embedded in the manuscript doesn’t contain the abbreviations and expressions that are explained in the figure legend

Line 381: ACT should be placed on Line 379

Line 402: VAS: name the full expression

Line 417: „type 2 targeted” dash or not dash?

Figure 4: the red area is impossible to decipher

Line 463: figure 5 legends include abbreviations which are not present in the image

Figure 7: the figure embedded in the manuscript doesn’t contain the abbreviations and expressions that are explained in the figure legend

Line 561: „(QoL)” delete

Line 561: „type 2 high” dash or not dash

Author Response

Dear Editor,

Dear Reviewers,

Thank you for your comments and suggestions. All of them guided us to ameliorate the article overall, providing additional details and better balancing contents, with particular reference to biological treatments.

We are also glad that the main objective of our work has been appreciated and that the proposed workflows have been considered helpful manual for physicians to assess patients with asthma and chronic rhinosinusitis with nasal polyps, define a personalized treatment, and select candidates for biological therapy as the multidisciplinary approach is essential for the appropriate management of the three different patient profiles that we have considered.

All authors contributed to the revision and to generate replays to reviewers’ observations.

REVIEWER 2

  1. This expert opinion is marked as a review article, however international guidelines reshaped to the local needs and opportunities should include personal observations to reinforce the consensus made by the contributors. Some of the self-cited articles demonstrate the authors’ experience, however, the materials and methods section does not highlight the research process. Whether it is a review (the process of article / guideline / literature screening e.g. PRISMA guidelines for systematic reviews and meta-analyses) or / and reasearch-based (local studies, observations, trials) expert opinion, a more detailed workflow of negotiation is welcome. The „two meetings” is a sort of vague and superficial description of the decision-making. On the other hand, the number of specialists in their field should be indicated.

Replay. More details regarding Italian Expert panel have been reported in paragraph 2 “Materials and Methods”.

  • Specialists were identified because of their broad experience in treating and managing severe asthma patients and comorbid CRSwNP, in the clinical practice. They also belong to Asthma Center where the multidisciplinary approach is already applied. In addition, the methodology regarding expert working groups and scientific literature selection has been better explained.
  • The article introduction has to be considered as a narrative review, therefore references have been selected by authors in order to provide a picture of asthma and CRSwNP, from different points of view (e.g., molecular mechanisms, medical needs, patient management approach); while workflows and related discussion are based on either authors’ opinion or scientific literature selected by the authors in order to support the argumentation of the three patient profiles.

  1. I ask the authors to provide a table containing all antibody-based medications available for asthma and CRSwNP, highlighting if a given drug is authorized for any or both conditions in the focus of this paper.

Nota Bene, that not all of the mentioned medications have a primary indication when CRSwNP is present. This is important when a patient’s assesment starts in the ENT office. It is confusing that, in the „Indications for biological treatment in CRSwNP” the medication list refers to those are available for asthma, however dupilumab is a medication that can be prescribed for both conditions as a primary choice of biological treatments. Dupilumab is not even mentioned the „Indications for biologics for patients with CRSwNP and asthma” section, where benrolizumab is overrepresented (ANDHI and OSTRO trials are also sponsored by AstrZeneca). This phenomenon raises concern when benrolizumab (Fasenra®), is a product of AstraZenca, which plays a funding role in this publication - as it is declared, AstraZeneca covered the copyediting expenses and is willing to support the publication expenses. This is also suspicious when the authors have nice real-life experiences with the competing drug (https://doi.org/10.3390/jcm11102684), but not cited in the current article. These issues must be solved by more transparent declarations, avoiding presenting results with a single drug. An expert opinion cannot be accepted when a medication is overrepresented (especially when it is not authorized as a biological agent for CRSwNP, but for asthma with an add-on effect – according to the recent negotiation between the FDA and AstraZeneca).

Replay. Authors appreciate this comment and agreed to change this part. The article was indeed updated in order to provide a complete picture of the five mAbs currently approved for severe uncontrolled asthma. See new Table 2, Line 117.
The paragraph (1.4 of the previous version) with more details regarding evidence from clinical trials or from real world setting has been delated; indeed, the aim of the article is not to explain treatment options, but to share patient management workflows based on a multidisciplinary approach.

  1. For all the observations regarding typos, copyediting, abbreviations, additional references (etc.)

Replay
. The whole content has been verified and corrected as needed.
Figures showing the workflow for each patient profile have been shifted at the beginning of each specific paragraph in order to facilitate the reading.
